# Combination of Lenvatinib and Pembrolizumab as Salvage Treatment for Paucicellular Variant of Anaplastic Thyroid Cancer: A Case Report

Cristina Luongo [1,*], Tommaso Porcelli [1] , Francesca Sessa [1], Maria Angela De Stefano [2], Francesco Scavuzzo [3], Vincenzo Damiano [2], Michele Klain [4], Claudio Bellevicine [1] , Elide Matano [2], Giancarlo Troncone [1] , Martin Schlumberger [5] and Domenico Salvatore [1]

1   Department of Public Health, University of Naples Federico II, 80131 Naples, Italy; tommaso.porcelli@unina.it (T.P.); sessafrancescaf@libero.it (F.S.); claudio.bellevicine@unina.it (C.B.); giancarlo.troncone@unina.it (G.T.); domenico.salvatore@unina.it (D.S.)
2   Department of Clinical Medicine and Surgery, University of Naples Federico II, 80131 Naples, Italy; mariaangela.destefano@unina.it (M.A.D.S.); vdamiano@unina.it (V.D.); ematano@unina.it (E.M.)
3   Department of Endocrinology, Aziena Ospedaliera di Rilievo Nazionale A. Cardarelli, 80131 Naples, Italy; scalen@alice.it
4   Department of Advanced Biomedical Sciences, University of Naples Federico II, 80131 Naples, Italy; micheleklain@libero.it
5   Department of Endocrine Oncology, Gustave Roussy, University Paris-Saclay, 94805 Villejuif, France; Martin.SCHLUMBERGER@gustaveroussy.fr
*   Correspondence: cristinaluongo@gmail.com

**Abstract:** Anaplastic thyroid cancer (ATC) is a rare but aggressive thyroid cancer, responsible for about 50% of all thyroid cancer-related deaths. During the last two decades, the development of a multimodal personalized approach resulted in an increased survival. Here, we present an unusual case of a 54-year old woman with a paucicellular metastatic ATC, a rare variant of ATC, who was treated with a combination of surgery, radiation therapy and cytotoxic chemotherapy. More than two years later, when the disease was rapidly growing, a combination of lenvatinib and pembrolizumab induced a partial tumor response of lung metastasis that persisted over 18 months. Paucicellular ATC may initially show a less aggressive behavior compared to other histological ATC variants. However, over the time, its clinical course can rapidly progress like common ATC. The combination of lenvatinib and pembrolizumab was effective as a salvage therapy for a long period of time.

**Keywords:** anaplastic thyroid cancer; immunotherapy; tyrosine kinase inhibitors

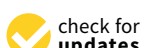



## 1. Introduction

Anaplastic thyroid carcinoma (ATC) is one of the most aggressive human solid tumors. ATC is a rare cancer, representing less than 2% of thyroid cancers, but it is responsible for half of thyroid cancer-related deaths. The median overall survival (OS) is 2–6 months [1–3].

Recently, the US Food and Drug Administration approved the combined treatment with dabrafenib and trametinib for ATC patients harboring the BRAF V600E mutation, which significantly improved the OS of this subset of patients (estimated 1-year OS 94%; hazard ratio 0.29 (95% CI, 0.10–0.78)) [1,4]. However, patients who have no targetable mutation still have a very poor prognosis. They are treated with a multimodal approach, consisting of surgical resection (whenever feasible), and external radiotherapy to the neck and mediastinum combined with chemotherapy [1,5]. This usually allows for an initial control of the local disease but not of distant metastases. A phase II study evaluated the efficacy of lenvatinib in combination with the anti-PD-L1 immune checkpoint inhibitor (ICI) pembrolizumab (an anti-PDL1 antibody) in patients with metastatic ATC or poorly differentiated thyroid carcinoma (PDTC) (NCT02973997). Among the six ATC patients,

four achieved a complete response (CR), the median PFS was 16.5 months, with three ATC patients still being alive without a relapse of the disease at 40, 27 and 19 months, respectively [6]. A differential diagnosis between ATC and PDTC could be insidious. Usually, ATCs have much higher degrees of cellular atypia and nuclear pleomorphism, a significantly higher mitotic activity, and confluent areas of tumor necrosis.

During the last decade, it has been emerging that ATC is a heterogeneous disease and that a more personalized approach could improve the patient prognosis. In this context, a recent French study showed that, among ATC patients treated with lenvatinib, the long-term survivors had a mixed histology of ATC combined with a more differentiated component [7]. Paucicellular ATC is an infrequent variant of ATC, which was associated with a better (although still poor) prognosis than classical ATC [8]. Here, we report the case of a 54-year old woman with a mixed epithelial-paucicellular variant of ATC that did not respond to two lines of chemotherapy but had a dramatic response to treatment with lenvatinib in combination with pembrolizumab.

## 2. Case Report

A 54-year old woman underwent a fine-needle aspiration biopsy (FNAB) for a 2.3 cm rapidly growing thyroid nodule (Figure 1). The cytological examination showed both solid groups and discohesive oxyphilic cells (Hürthle cells) in a background featuring lymphocytes. Based on these features, the FNAB was diagnosed as a low-risk indeterminate lesion (AUS/FLUS). Five months later, the nodule grew to 3.6 cm, and thus another FNAB was performed; a diagnosis of suspicious for malignancy was rendered. The patient underwent a total thyroidectomy (nodule 4.1 × 3.4 cm) with cervical lymph node dissection, and a removal of the internal right jugular vein that was invaded by the tumor. Microscopically, a Hürthle cell carcinoma with foci of paucicellular anaplastic cancer was diagnosed (Stage IVB; cT3b cN0 Mx/pT4b pN0 M0). In particular, large epithelial cells featuring granular eosinophilic cytoplasms, hyperchromatic nuclei with evident nucleoli were arranged in a solid and trabecular pattern alternated with scattered anaplastic spindle cells and necrotic areas. Immunohistochemical stainings for pancytokeratin and PAX8 were positive in both these components. Conversely, TTF1 was expressed by Hürthle cells only. Thyroglobulin (Tg) immunostaining was negative in both Hürthle and anaplastic spindle cells (Figure 2).

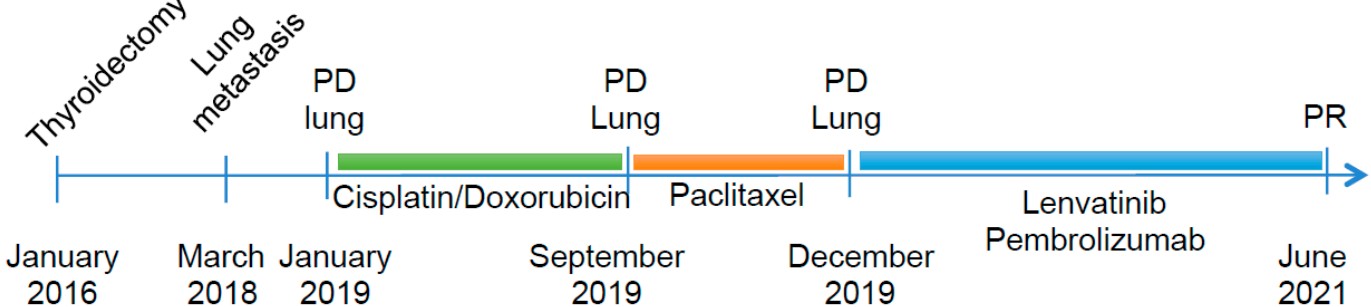

**Figure 1.** Schematic representation of disease progression and treatments; PD: progression disease, PR: partial response.

Two years later, because of the appearance of a hacking cough, a 18-fluorodeoxyglucose (18-FDG) positron emission tomography (PET) scan was performed and revealed several millimetric lung hypermetabolic areas. Over time, the serum Tg under LT4-suppressive therapy had increased from 0.15 to 19 ng/mL. Two months later, a computed tomography (CT) scan revealed the presence of multiple lung lesions, in particular one in the medium lobe invading the airways (21 mm diameter), a second in the right inferior lobe (4 mm diameter) and a third in the left lung (9 mm diameter). A transbronchial biopsy of the largest lesion was performed, and the histology was consistent with a thyroid cancer metastasis.

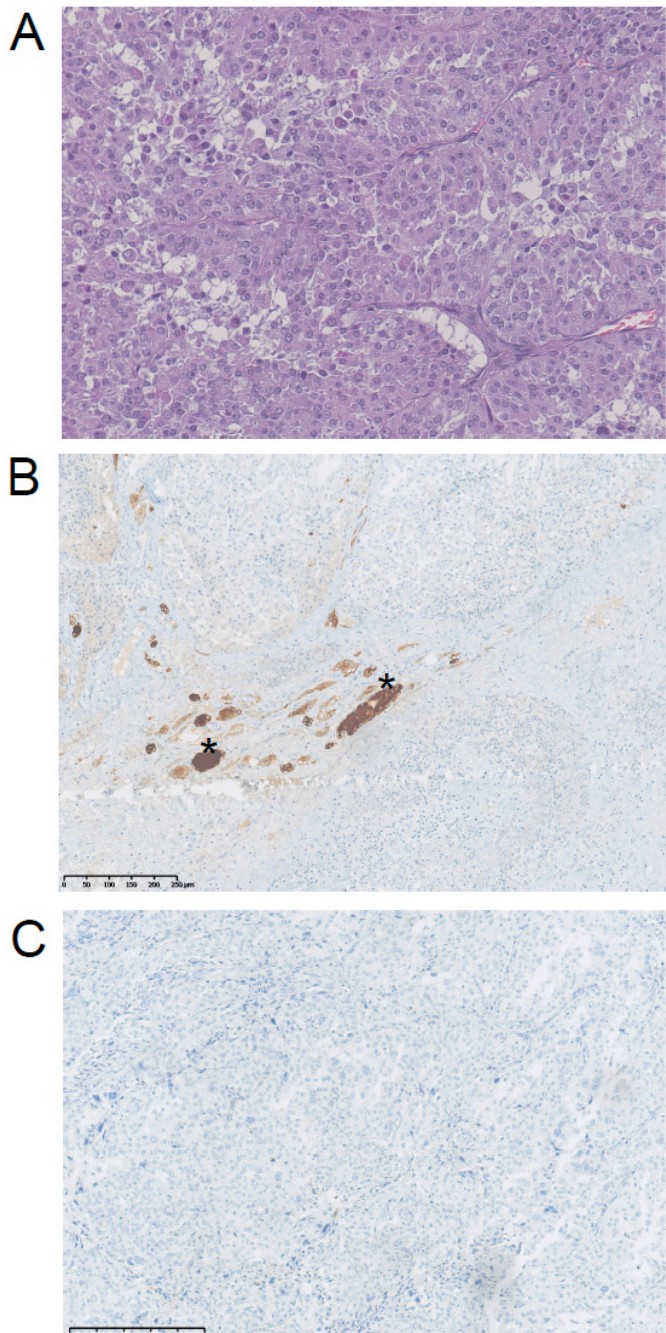

**Figure 2.** Immunohistochemical staining of the ATC primary tumor that developed in the context of a Hürthle cell carcinoma. (**A**) H&E staining of ATC with epithelial-paucicellular pattern (40×), (**B**) Tg staining was negative; positively stained follicles (stars) were normal cells entrapped by the tumor cells (internal control); (10×) (**C**) PD-L1 staining was negative (10×).

On January 2019, the patient came to our observation, 18FDG-PET/CT showed a further increase in the number, size and FDG uptake of the lung lesions, and the appearance of multiple lymphadenopathies in the neck and mediastinum. In particular, the lesion in the right inferior lobe had grown to 8.5 cm, with a SUV max of 50. The histology of the tumor has been reviewed by two independent pathologists who confirmed the diagnosis of ATC. The molecular test performed on the primary thyroid tumor for BRAF, NTRK and ALK mutations was negative, and chemotherapy with cisplatin plus doxorubicin was given. After three courses of treatment the disease was stable, but after three further courses the disease progressed. The patient was then given four courses of paclitaxel;

however, the disease continued to progress. Immunohistochemistry for PD-L1 performed on the primary tumor was negative, but the presence of tumor-infiltrating lymphocytes (TIL) was noticed. The primary tumor was also tested for microsatellite instability-MSI using a panel of five markers (BAT25, BAT26, D2S123, D5S346 and D17S2720), and the tumor was instable for the presence of mutations of BAT-26, D2S123 and D17S2720. Based on these results, on January 2020 a treatment with lenvatinib (24 mg daily) in combination with pembrolizumab (200 mg every 21 days) was started as a salvage therapy. After five months of treatment, a partial response was achieved with a reduction by 76% of the lung lesion in the medium lobe from 8.5 cm to 2 cm and at 18FDG PET/CT a drop in SUVmax from 50 to 4.7. The treatment was continued, but lenvatinib was progressively reduced to 10 mg daily due to grade 3 diarrhea, vomiting and weight loss. After 12 months, the administration of pembrolizumab was delayed every six weeks. The lenvatinib treatment has been withdrawn several times, and the patient compliance was low. In spite of that, after 18 months of treatment all lung lesions continued to respond to therapy (PR with a reduction by 50% of the sum of the diameters of the target lesions by RECIST) (Figure 3). Unfortunately, a pleural nontarget lesion showed a progression, over four months, as demonstrated by the 18FDG-PET/CT (10 mm vs. 7 mm, a 43% increase by RECIST and a SUV max of 25.8 vs. 6.5) (Figure 1). The pleural lesion is being treated with stereotactic radiotherapy (42 Gy in seven fractions). The patient is still alive, and the treatment with lenvatinib and pembrolizumab is still ongoing.

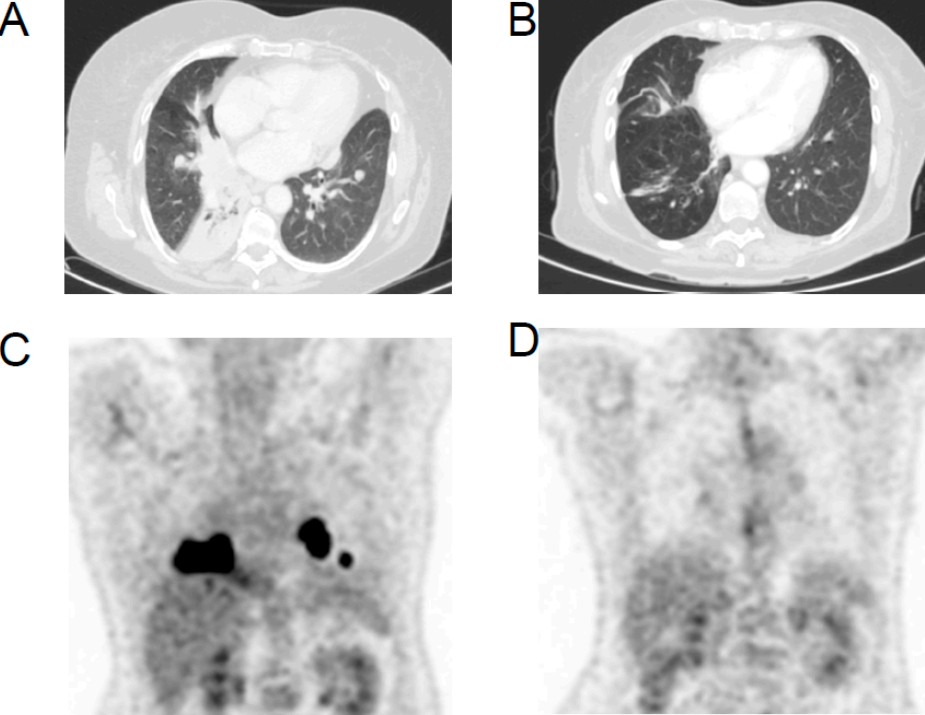

**Figure 3.** Axial CT image of lung (**A**) lesion before the start of combined lenvatinib/pembrolizumab treatment and (**B**) after 18 months of treatment; representative 18FDG-PET/CT images (**C**) at the initiation of the combined lenvatinib/pembrolizumab treatment and (**D**) after 18 months of treatment.

## 3. Discussion

We describe a case of a 54-year old woman with a rare ATC variant that developed in the context of a Hürthle cell carcinoma and was treated with a combination of pembrolizumab plus lenvatinib as a salvage therapy after surgery and two lines of chemotherapy. Concerning this case, there are several points to discuss.

Only few cases of paucicellular variants of ATC have been reported [8–11]. This case confirmed that the age of presentation was younger than conventional ATC, which

usually occurs in patients older than 55 years. The paucicellular variant of ATC might be misdiagnosed as subacute thyroiditis due to the low cellularity, spindle cells and multinucleated giant cells and lymphocytic infiltration within the tumor [10,11]. Another interesting aspect is that this ATC variant is associated with a longer survival; Canos et al. described two cases of paucicellular ATC without evidence of disease, 11 and 15 months after surgery, respectively [8]. Our patient was free of disease for 26 months after the surgery. However, when lung metastases appeared, the behavior of the disease was very similar to the conventional ATC, with a rapid growth of multiple metastases in the lungs and lymph nodes and a resistance to chemotherapies. Due to the unusual long PFS after the surgery, the histological revision of the tumor has been mandatory. The more difficult differential diagnosis was with the PDTC. The percentages of cellular atypia and nuclear pleo-morphism are higher in ATCs than PDTC, as is the cellular proliferation rate measured by Ki-67 percent positive tumor nuclei ($\geq$30%). Furthermore, ATCs have confluent areas of tumor necrosis, and the thyroglobulin expression is absent, whereas PDTC are generally characterized by single cells or focal patches of tumor necrosis, and thyroglobulin expression is retained.

The therapeutic options for patient with ATC without a targetable mutation are still limited. Beyond the multimodal approach with surgery, radiotherapy and chemotherapy, it has been proposed that one treat these patients with TKIs alone or in combination with immunotherapy [6,12–14]. Several TKIs, such as pazopanib [15], axitinib [16] and sorafenib [17], were not effective when used alone for the treatment of ATC. A prospective multicentric phase II trial did not confirm the encouraging results of a Japanese phase II study of lenvatinib in ATC patients and was interrupted due to the absence of efficacy [18,19]. Recent studies suggested that one start the treatment with lenvatinib alone and add pembrolizumab upon disease progression [4,20–22]. However, some TKIs induce modifications of the tumor environment that inhibit the anti-tumor immune response, thus contributing to immune resistance and progression of disease [23]. Therefore, it has been suggested that a combination of immunotherapy and TKI should be used from treatment initiation. Moreover, in some tumors, the hypoxia determined by VEGF inhibition increases PD-L1 expression, synergizing with anti-PD1 therapy [24,25]. Ivery et al. showed that a median PFS of 2.96 months was attained with the addition of pembrolizumab in patients already resistant to TKI treatment, compared to a median PFS of 16.5 months in patients treated with lenvatinib and pembrolizumab from the beginning [14]. In the present case, the combined treatment with pembrolizumab and lenvatinib was started simultaneously and was effective, although the PD-L1 expression in the primary tumor was negative. However, the PD-L1 expression status in the metastasis that developed two years after the diagnosis of the thyroid tumor could not be assessed. On the other hand, a lymphocyte infiltration of the tumor and a microsatellite instability were evidenced, two characteristics that are associated with tumor response to immunotherapy [26–29]. Unfortunately, we have not been able to quantify the tumor mutation burden, due to the lack of tissue material, and therefore we cannot exclude that metastases arose from the Hürthle cell component, despite the rapid growth being in favor of an anaplastic origin. The partial response was prolonged for 18 months, after which a single nontarget lesion progressed and was amenable to focal radiation therapy. This tumor progression could be related to an acquired resistance of one lesion or to the decreased doses of lenvatinib and the poor compliance of the patient.

In conclusion, this case confirmed that the behavior of metastatic paucicellular ATC might be very aggressive and poorly responsive to conventional chemotherapy. The combined treatment with pembrolizumab and lenvatinib might represent an effective salvage therapy for these patients.

**Author Contributions:** Conceptualization, C.L. and D.S.; methodology, E.M., V.D., C.B. and G.T.; formal analysis, C.L. and M.K.; investigation, T.P., F.S. (Francesca Sessa), F.S. (Francesco Scavuzzo) and M.A.D.S.; resources, D.S.; data curation, C.L.; writing—original draft preparation, C.L. and M.S.; writing—review and editing, C.L., D.S. and M.S.; supervision, D.S. and M.S. All authors have read and agreed to the published version of the manuscript.

**Funding:** This research received no external funding.

**Institutional Review Board Statement:** Ethical review and approval were waived for this study, due to off-label therapy administration not requiring the submission of any request to our ethics committee.

**Informed Consent Statement:** Written informed consent has been obtained from the patient to publish this paper.

**Conflicts of Interest:** The authors declare no conflict of interest.

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
