# Peer review of "Combination of Lenvatinib and Pembrolizumab as Salvage Treatment for Paucicellular Variant of Anaplastic Thyroid Cancer: A Case Report"

_curroncol, doi:10.3390/curroncol28060450_

Round 1
Reviewer 1 Report
This is a single case report. As much as we want to read into it, we have only limited ability to do so:
This case report details the successful treatment of pt with rare anaplastic thyroid cancer (ATC) with lenvatinib and pembrolizumab.
The treatment is relevant because while lenvatinib is approved for DTC, the use of this single agent in ATC has limited efficacy with extremely short DFI and variable ORR. Pembrolizumab is a PD-1 checkpoint inhibitor which is classically useful in high PD-L1 expressed Ca lung. This combination of lenvatinib and pembrolizumab has been shown to be effective in small series, most recently published in Thyroid journal by Dierks et al. In that series they reported 4/6 (66%) complete remission rate, with a median PFS of 17.75 months.
This single case report adds to the growing body of case reports and case series of the efficacy of this combination in ATC.
The authors have done an adequate job describing and summarising their single-patient experience with this combination.
This is a coherently written case report supported by the data presented.
There are several syntax and grammatical errors that are present and should be addressed. Otherwise this is a relevant case report for the treatment of anaplastic thyroid Ca.
Author Response
We thank the reviewer for the comments. The syntax and grammatical errors have been corrected.Reviewer 2 Report
Dear authors,
I thank you for your manuscript. Please find my comments hereunder:
1) Line 18: cytotoxic chemotherapy: is there a non-cytotoxic chemotherapy? if no please delete cytotoxic here and further in text.
2) line 30: US Food and Drug Administration.
3) line 32: It is good informative to say how is the improvement (so without treatment 2-6 months as you mentioned in line 28 and how is that with BRAF-inhibitors?
4) Line 37-38: what about BRAF-inhibitors as you mentioned before in these = metastatic patients?
5) Line 37-38: you say about using TKI in metastatic patients as an alternative treatment. If it is an alternative treatment what is the other one?
6) Line 61: you present nicely your case. It would be nice to explain a bit more about paucicellular variant (clinical and pathological features).
7) Line 70: it is then pT4bpN0, would you please report your clinical cTNM staging
8) Line 71: would you please explain why patient have not received an adjuvant treatment after surgery?
9) Line 86-93: please explain the rational for your wait and see / conservative approach between March 2018 and January 2019.
10) Line 108: would you explain your consideration to choose the combination of Lenvatinib and Pembrolizumab and not Lenvatinib alone?
11) Line 112: The administration dose of Lenvatinib was reduce due to grade 3 diarrhea, vomiting and weight loss. Why weight loss was important for you regarding using Lenvatinib?
12) Line 118: would you please, if possible, report the radiation dose and schedule?
Author Response
We thank the reviewer for the interesting comments.
1) Line 18: cytotoxic chemotherapy: is there a non-cytotoxic chemotherapy? if no please delete cytotoxic here and further in text.
Cytotoxic has been deleted from the text as suggested.
2) line 30: US Food and Drug Administration.
The name of the American organization has been modified as suggested.
3) line 32: It is good informative to say how is the improvement (so without treatment 2-6 months as you mentioned in line 28 and how is that with BRAF-inhibitors?
This information has been added to the manuscript.
4) Line 37-38: what about BRAF-inhibitors as you mentioned before in these = metastatic patients?
Starting from line 32, we refer to patients with ATC without any targetable mutations, for this reason we did not include this information in the main text.
5) Line 37-38: you say about using TKI in metastatic patients as an alternative treatment. If it is an alternative treatment what is the other one?
TKI in metastatic patients is an alternative treatment to chemotherapy (which was the only therapeutic option until few years ago).
6) Line 61: you present nicely your case. It would be nice to explain a bit more about paucicellular variant (clinical and pathological features).
We agree with the reviewer and have discussed clinical and pathological features of the ATC paucicellular variant in the discussion section (please see lines 165-175).
7) Line 70: it is then pT4bpN0, would you please report your clinical cTNM staging
This information has been added to the manuscript (line 71)
8) Line 71: would you please explain why patient have not received an adjuvant treatment after surgery?
The patient came to our observation on January 2019 (line 114). Therefore, we do not know why the previous endocrinologist chose not to give adjuvant treatment to the patient
9) Line 86-93: please explain the rational for your wait and see / conservative approach between March 2018 and January 2019.
As mentioned before, unfortunately, the patient came to our observation only on January 2019 (line 114). We do not know why the previous endocrinologist had that conservative approach.
10) Line 108: would you explain your consideration to choose the combination of Lenvatinib and Pembrolizumab and not Lenvatinib alone?
The reviewer may now find the answer to this question in the discussion section, lines 184-188.
11) Line 112: The administration dose of Lenvatinib was reduce due to grade 3 diarrhea, vomiting and weight loss. Why weight loss was important for you regarding using Lenvatinib?
Because the cachexia (a loss of more than 5 percent of body weight over 12 months or less) could have a negative impact on the survival.
12) Line 118: would you please, if possible, report the radiation dose and schedule?
This information has now been added to the manuscript.
Reviewer 3 Report
The manuscript presents the reassuring result of the combination therapy of lenvatinib and pembrolizumab in an advanced form of paucicellular anaplastic cancer. The review of the literature is appropriate, the rare variant of ATC is interesting for the scientific community. Despite the current development of targeted therapies, the management of ATC is still extremely difficult and hopeless in the majority of cases. Trials using the combinations of TKI and immunotherapy are in progress. This case report confirms that it is useful to try the combination treatment while we wait for the results of ongoing trials and the individual approach is reasonable.
Author Response
We thank the reviewer for the comments